# Speed-dependent changes in the arm swing during independent walking in individuals after stroke

Daan De Vlieger[1,2,3☉], Arne Defour[1,2,3☉], Lynn Bar-On[1,3], Dirk Cambier[1,3], Eva Swinnen[2,3,4,5], Ruth Van der Looven[6], Anke Van Bladel[1,3,6]*

1 Department of Rehabilitation Sciences, Ghent University, Ghent, Belgium, 2 Rehabilitation Research Group (RERE), Vrije Universiteit Brussel, Laarbeeklaan, Brussel, 3 Rehabilitation Technology for Persons with a Brain Injury Alliance Research Group (REBI), Belgium, 4 Brussels Human Robotics Research Center (Brubotics), Vrije Universiteit Brussel, Laarbeeklaan, Brussel, 5 Center for Neurosciences (C4N), Vrije Universiteit Brussel, Laarbeeklaan, Brussel, 6 Department of Physical and Rehabilitation Medicine, Ghent University Hospital, Ghent, Belgium

☉ These authors contributed equally to this work.
* Anke.VanBladel@UGent.be

**Data Availability Statement:** All relevant data are within the manuscript and its Supporting information files.

## Abstract

### Background

Increasing one's walking speed is an important goal in post-stroke gait rehabilitation. Insufficient arm swing in people post-stroke might limit their ability to propel the body forward and increase walking speed.

### Purpose

To investigate the speed-dependent changes (and their contributing factors) in the arm swing of persons post-stroke.

### Material and methods

Twenty-five persons post-stroke (53±12.1 years; 40.72±43.0 months post-stroke) walked on a treadmill at comfortable (0.83m/s) and fast (1.01m/s) speed. Shoulder and elbow kinematics were compared between conditions using Statistical Parametric Mapping (SPM) analysis, and discrete parameters using a Wilcoxon signed-rank test or an independent sample t-test. The relations between speed-dependent changes in shoulder and elbow range of motion and clinical and gait parameters were assessed using Spearman correlation coefficients.

### Results

The non-paretic arm showed expected speed-dependent kinematic adaptations with increases in active range of motion for shoulder flexion (p<0.001), extension (p<0.05), abduction (p = 0.001), rotation (p = 0.004) and elbow flexion (p<0.001). The paretic arm only showed an increase in shoulder abduction and elbow flexion (both p<0.001). Persons post-stroke with a more impaired arm swing coordination pattern only showed speed-dependent

**Funding:** Published with support of the Universitaire Stichting from Belgium and of the Rehabilitation technology for persons with a Brain Injury alliance research group.

**Competing interests:** The authors have declared that no competing interests exist.

adaptations for elbow flexion (p<0.001) at the paretic side during fast walking. In contrast, persons post-stroke with a normal arm swing coordination pattern presented with increases in active range of motion of the shoulder abduction and elbow flexion (both p<0.001) at the paretic side when walking fast. More upper limb impairment (r = -0.521, p<0.01) and a wider step width (r = 0.534, p<0.01) were related to a larger increase in mean elbow flexion during faster walking.

## Conclusions

Persons post-stroke show different changes in arm swing kinematics at the paretic compared to the non-paretic side when increasing walking speed. The changes are related to the impairment level and stability during walking, indicating that therapeutic interventions aiming to increase walking speed by improving arm swing might need to target these factors.

## Introduction

Most persons post-stroke regain independent walking to a certain extent, but remaining gait impairments have a great impact on one's functional performance, independence and quality of life [1]. Slower walking speed is linked to a more impaired gait [2] and increased fall risk [3] while an increase in walking speed over time is associated with an increase in quality of life [4]. Therefore, increasing walking speed is often an important goal in post-stroke gait rehabilitation.

While in persons post-stroke increasing the step length bilaterally helps to increase walking speed [5], in healthy subjects, speed-dependent adaptations of the upper-limbs also contribute to increasing walking speed. Specifically, increases in range of motion (ROM) of the elbow flexion, shoulder flexion and shoulder abduction, proportional with the increased step length, help to accelerate the body forwards [6, 7]. This increased ROM is probably initiated by an increased activity of the arm muscles during walking [8] and is accompanied by a shift of the peak shoulder flexion moment to a later point in the gait cycle [7]. This implies concurrent speed-dependent changes in interlimb coordination (i.e. coordination between arms and legs) [9].

As a direct result of the stroke or as an adaptive strategy, persons post-stroke often move the affected arm less than the non-affected arm during walking [10, 11]. It is expected that during walking at a self-selected speed, the non-affected arm of a person post-stroke follows a healthy, reciprocal 1:1 coordination pattern (i.e. one arm moves in the same direction as the contralateral leg during one stride). In contrast, the affected arm shows an altered interlimb coordination [8, 12, 13]. While in healthy adults a 2:1 arm-to-leg swing ratio (two arm swings, one stride) can occur when walking slower than the comfortable walking speed, persons post-stroke sometimes show a 2:1 ratio on the paretic side independent of walking speed. Wagenaar et al. (1994) previously described these two arm-swing coordination patterns [9], which were recently confirmed by Van Bladel et al. (2023) [14].

Only a handful of studies have investigated the relationship between arm swing and walking speed in persons post-stroke. Stephenson (2009) reported an increase in arm swing amplitude at the non-paretic side, while the frequency of the arm swing movements increased bilaterally [11]. However, since participants in this study could support on sliding handles, these results

cannot be generalized to free arm swing movements. Bovonsunthonchai (2012) described improved coordination between the affected upper limb and the unaffected lower limb in persons post-stroke with increasing walking speed but, not between the unaffected upper limb and the affected lower limb [15]. In contrast, Stephenson (2009) did not report changes in the interlimb coordination at a higher walking speed compared to comfortable walking [11]. Finally, an indirect relationship between walking speed and arm movements has been described when treatment with botulinum toxin A injections, to reduce upper limb muscle spasticity, was shown to have a positive effect on stride time and walking speed [16–18]. Despite these available indications of a possible relationship between arm swing and walking speed in persons post-stroke, so far no study assessed the changes in free arm movements during fast walking.

Knowing whether persons post-stroke are capable of adapting their arm movements, and which factors are associated with this ability, may be a first step in aiding post-stroke gait rehabilitation as a potential way to contribute to walking speed. Hereby the person post-stroke's ability to walk safely in the community increases which is critical for the preservation of their independence, their social integration and their participation in life roles [19–21]. Therefore, the first aim of this cross-sectional study was to examine if independent walking persons post-stroke show speed-dependent changes in upper limb kinematics during treadmill walking. The second aim was to examine if the speed-dependent changes of the upper limbs are different depending on the coordination pattern of the paretic arm (arm-to-leg swing ratio) during comfortable walking [14]. Since there was a clear difference in upper limb impairment (Fugl-Meyer assessment for the upper limb, FMUL) between these groups, it is expected that they react differently to an increase in walking speed. The third aim was to investigate the relationship between the change in upper limb ROM and upper/lower limb impairment, the increase in walking speed, step width during comfortable and fast walking and spasticity of shoulder internal rotators and elbow flexors to gain insight in which factors contribute to speed-dependent changes in arm swing. We hypothesized that an increase in upper limb ROM can be identified with increasing walking speed but that participants with an altered arm-to-leg swing ratio and more impaired clinical parameters will have less potential to adapt their arm swing movement.

## Materials and methods

### Participants

Participants were recruited from the rehabilitation center of the Ghent University Hospital and through advertising on social media. Persons post-stroke could participate if they could walk independently (i.e. without walking aids) for at least six minutes as indicated by a Functional Ambulation Categories score ≥ 3 [22]. Participants were excluded if they suffered from other neurologic, musculoskeletal, respiratory or severe cardiovascular disorders that affected gait performance. Other exclusion criteria were a bilateral, cerebellar or recurrent stroke, lower limb orthopedic surgery in the past year and cognitive or language impairments that hamper the patients from understanding simple tasks.

This study was approved by the Medical Ethics Committee of the Ghent University Hospital (B670201941390) and registered in a public repository (NCT 04180124; 21/10/2019). All recruited participants agreed and signed written informed consent prior to the study. Participants were recruited between 28/1/2020 and 5/9/2020.

### Study protocol

Several clinical parameters were used to describe the participants impairments. Passive ROM (degrees, °) for shoulder flexion, abduction, external rotation and elbow flexion of the paretic

arm was measured using a goniometer [23]. The motor function of the paretic limbs was quantified using the Fugl-Meyer assessment for the upper and lower limb (FMUL and FMLL with scores ranging 0–66 and 0–34 respectively) [24]. Spasticity of shoulder (extensors, internal rotators, adductors) and elbow (flexors, extensors) muscles was assessed using the Modified Ashworth Scale (scores ranging from 0–4) [25].

Subjects walked on an instrumented treadmill (R-Mill Forcelink, The Netherlands) in a Virtual Reality environment (GRAIL, Motek Medical BV, The Netherlands) wearing a non-weight-bearing safety harness (JSP, PN 21) and without holding the handrails. After six minutes of familiarization, all participants completed two 3-minute trials at comfortable walking speed (CWS) and at a fast walking speed (FWS). The last two minutes were recorded and used for data-analysis. CWS was determined by increasing the average walking speed of the familiarization trial with increments of 0.1 m/s. Participants indicated their CWS before data recording started. The FWS trial was determined by increasing the CWS (increments of 0.01m/s) until a level that was still feasible as verbally indicated by the participant. The difference in walking speed between both conditions was at least 0.10m/s [26]. Subjects were allowed to rest between trials however, none of the participants expressed the need to rest.

## Data processing

Three-dimensional motion data was captured (Vicon Motion Systems, Oxford, UK) at 100Hz using the Full Body Plug-In Gait model [27]. Data were low-pass filtered (bidirectional 4th-order Butterworth filter). Marker labelling, stride event detection (based on vertical ground reaction forces with a threshold of 20N) and kinematic calculations were performed in Nexus (version 2.9.3). Visual3D (v6.01.36, C-motion Inc., USA) was used to remove strides with incorrect foot placement (i.e. crossing the midline with landing (partially) on the force plates of the opposite side) and to export time-normalized data (101 data frames). To define arm swing, kinematic curves of the shoulder in the three planes and the elbow in the sagittal plane, averaged over at least 17 time-normalized strides, were extracted. These four kinematic variables were selected due to their significant contribution to the arm swing motion rather than elbow pro- and supination or wrist movements. To control for potential compensations, kinematic curves of the trunk (three planes) were also extracted. Per subject, and per walking speed condition the average ROM (difference between the maximum and minimum) of the shoulder movements in each plane and elbow movement in the sagittal plane were defined. In addition, the average elbow flexion per speed was determined. Gait parameters included the difference in walking speed and step width (as a measure of gait stability) between CWS and FWS. All the discrete parameters were averaged over the time-normalized strides.

## Statistical analysis

Statistical Parametric Mapping (SPM) was conducted (www.spm1d.org) running an SPM (t) script to compare the time-normalized kinematic curves between the two conditions by calculating the conventional univariate t-statistic at each point of the gait cycle. To estimate the magnitude of significant differences between the two conditions, mean difference curves were calculated. Discrete parameters were analyzed using SPSS statistics for Windows Version 28. Normality of the data was verified using the Shapiro-Wilk test. Accordingly, a Wilcoxon signed-rank test was used to compare conditions and an independent sample t-test was used to compare subgroups. Spearman correlation coefficients were calculated between speed-dependent changes in arm swing kinematics (ROM of shoulder and elbow movements and

mean elbow flexion) and the clinical (FMLL, FMUL) and gait (changes in gait-speed, step width) related parameters. The strength of the correlations was classified as very strong (r≥0.80), strong (0.60–0.79), moderate (0.40–0.59) or weak (0.20–0.39) [28]. Statistical significance was set at p<0.05.

Subgroup analyses were performed based on different arm swing coordination patterns during the CWS condition. These subgroups were defined using Fast Fourier transform (FFT) analysis (Matlab R2020b) performed on the kinematic data [29] of the hemiplegic shoulder movements in the sagittal plane averaged over the strides as the pattern was consistent throughout strides. The frequency of the arm swing was determined from the single-sided amplitude spectrum plots of the FFT analysis that assessed whether there was a dominant frequency of 1 or 2 shoulder movements per stride. Several shoulder flexion-extension movements may be present during one stride. However, we determined that when the amplitude spectrum of an arm swing frequency of 2 was higher than half of the amplitude spectrum of an arm swing frequency of 1, the subject was categorized as having a 2:1 pattern, and when less than half, as having a 1:1 pattern (Fig 1). One participant showed a reversed arm swing pattern (i.e. ipsilateral arm and leg move together in the same direction) and was therefore excluded from the subgroup analysis. The researchers suggest that the reversed arm swing pattern could potentially be explained by altered cognitive functioning of the patient.

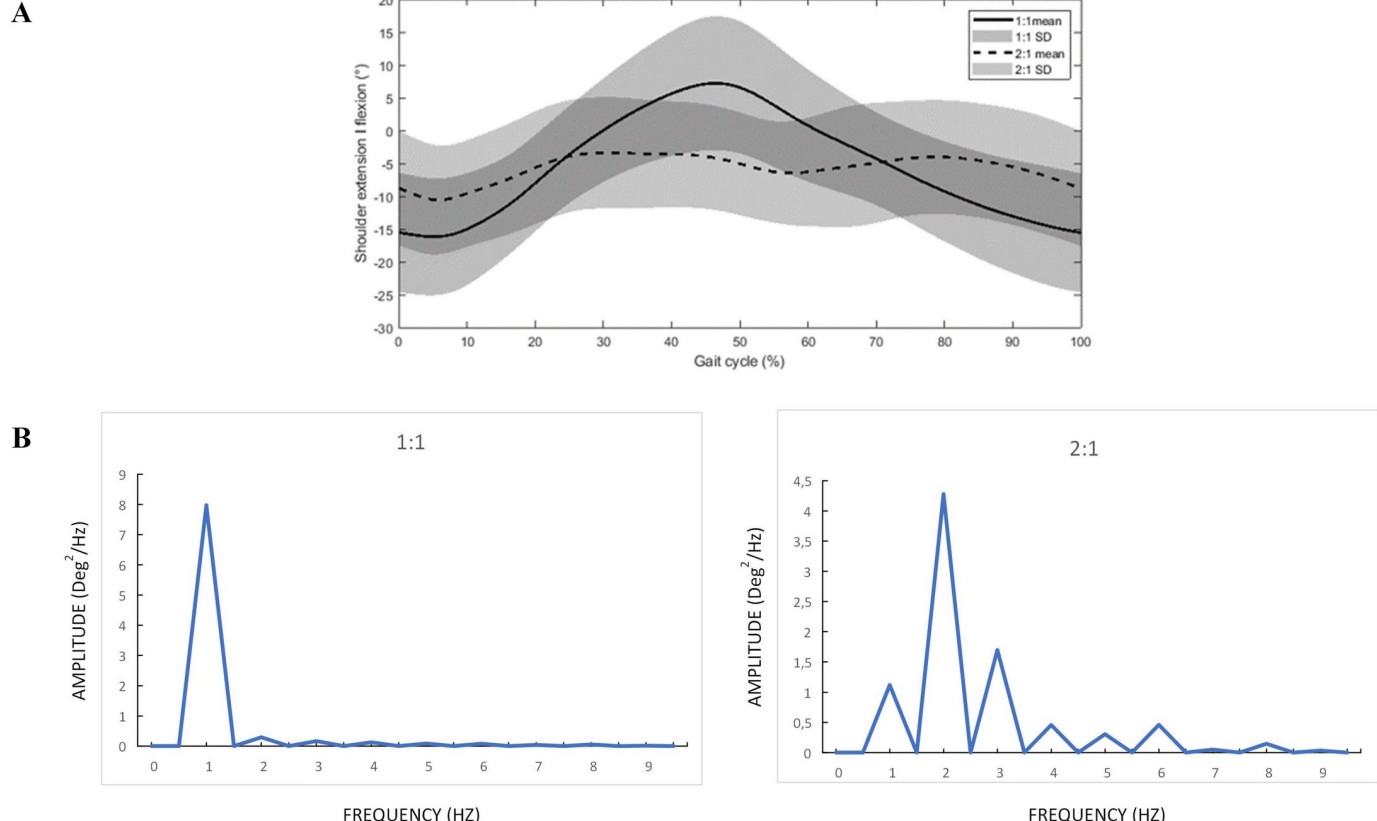

**Fig 1. Graphical representation of the sagittal shoulder movements of the subgroups and examples of accompanying amplitude spectrum plots.** (A) Sagittal shoulder movements of the subgroups, (B) Amplitude spectrum plots.

**Table 1. Demographics and clinical parameters for the entire group and per subgroup.**

| | Full group (N = 25) | | | 1:1 subgroup (N = 15) | | | 2:1 subgroup (N = 9) | | | P 1:1 versus 2:1 |
|---|---|---|---|---|---|---|---|---|---|---|
| | Mean (SD) | Range | N | Mean (SD) | Range | N | Mean (SD) | Range | N | |
| **Age** (y) | 53.00 (12.1) | 26–68 | | 56.67 (10.5) | 29–68 | | 49.00 (12.5) | 26–63 | | **0.041*** |
| **Sex** (F/M) | | | 10/15 | | | 5/10 | | | 4/5 | |
| **Time post-stroke** (m) | 40.72 (43.0) | 2–168 | | 35.40 (34.3) | 2–108 | | 53.78 (55.4) | 5–168 | | 0.323£ |
| **Stroke type** (I/H) | | | 15/10 | | | 10/5 | | | 5/4 | |
| **Paretic side** (L/R) | | | 12/13 | | | 8/7 | | | 4/5 | |
| **CWS** (m/s) | 0.83 (0.2) | 0.43–1.40 | | 0.86 (0.2) | 0.43–1.40 | | 0.81 (0.2) | 0.60–1.15 | | 0.641£ |
| **FWS** (m/s) | 1.01 (0.2) | 0.53–1.60 | | 1.04 (0.3) | 0.53–1.60 | | 0.99 (0.2) | 0.70–1.35 | | 0.593£ |
| **Δ Walking speed** (m/s) | 0.18 (0.04) | 0.10–0.25 | | 0.19 (0.04) | 0.10–0.25 | | 0.17 (0.04) | 0.10–0.22 | | 0.640* |
| **Step width CWS (cm)** | 0.21 (0.04) | 0.11–0.30 | | 0.20 (0.04) | 0.11–0.30 | | 0.21 (0.03) | 0.17–0.27 | | 0.755£ |
| **Step width FWS (cm)** | 0.21 (0.04) | 0.14–0.29 | | 0.21 (0.04) | 0.14–0.29 | | 0.21 (0.03) | 0.19–0.27 | | 0.710£ |
| **FMLL** (/34) | 28.12 (5.0) | 12–34 | | 29.80 (3.4) | 20–34 | | 25.44 (6.3) | 12–33 | | **0.041*** |
| **FMUL** (/66) | 47.48 (17.4) | 15–66 | | 56.07 (12.1) | 21–66 | | 32.22 (15.5) | 15–54 | | **<0.001*** |
| **FAC** (3/4/5) | | | 1/9/15 | | | 1/4/10 | | | 0/4/5 | |
| **AFO** (n) | | | 6 | | | 2 | | | 4 | |
| **PARETIC UL PROM** (°) (**) | | | | | | | | | | |
| Sh flexion | 155.60 (24.0) | 90–180 | | 157.33 (21.9) | 90–180 | | 151.11 (28.9) | 90–180 | | |
| Sh abduction | 141.00 (26.2) | 90–170 | | 143.33 (23.2) | 90–170 | | 135.00 (32.0) | 90–170 | | |
| Sh external rotation | 45.80 (29.6) | 0–90 | | 48.33 (30.6) | 5–90 | | 40.00 (30.3) | 0–80 | | |
| **MAS** (0/1/1+/2/3/4) | | | | | | | | | | |
| Sh retroflexors | | | 21/2/2/0/0/0 | | | 15/0/0/0/0/0 | | | 5/2/2/0/0/0 | | |
| Sh adductors | | | 21/3/1/0/0/0 | | | 15/0/0/0/0/0 | | | 5/3/1/0/0/0 | | |
| Sh internal rotators | | | 15/5/4/1/0/0 | | | 12/2/1/0/0/0 | | | 2/3/3/1/0/0 | | |
| El flexors | | | 9/14/2/0/0/0 | | | 7/8/0/0/0/0 | | | 1/6/2/0/0/0 | | |
| El extensors | | | 14/5/4/2/0/0 | | | 12/3/0/0/0/0 | | | 1/2/4/2/0/0 | | |

SD = standard deviation; N = number; Y = years; F = female; M = male; m = months; I = ischemic; H = hemorrhagic; L = left; R = right; CWS = comfortable walking speed; FWS = fast walking speed; Δ Walking speed = difference in walking speed between CWS and FWS; FMLL = Fugl-Meyer Assessment lower limb; FMUL = Fugl-Meyer Assessment upper limb; FAC = Functional Ambulation Categories; AFO = ankle foot orthosis; UL = upper limb; PROM = passive range of motion;

MAS = Modified Ashworth Scale, Sh = shoulder; El = elbow,

£ Independent sample T-test,

* Mann-Whitney U test

(**) All participants had full passive range of motion for elbow flexion and extension

## Results

Twenty-five persons post-stroke participated (10 females/15 males; 53 ±12.1 years; 40.72 ± 43.0 months post-stroke). Based on the Fugl-Meyer scores, participants showed a mild impairment of the lower limb and a moderate to mild impairment of the upper limb [30]. Demographics and clinical parameters for the entire group and subgroups are presented in Table 1.

### Speed-dependent changes in arm swing for the entire group

The kinematic curves of the paretic and non-paretic upper limb of both conditions are presented in Fig 2. Mean difference curves (with SD) of the movements where there were significant differences can be found in S1 Fig in S1 File. SPM analyses indicated that at the paretic side, persons post-stroke showed greater shoulder abduction (p<0.001) and elbow flexion (p<0.001) in the FWS condition over the entire gait cycle (GC). In the sagittal plane,

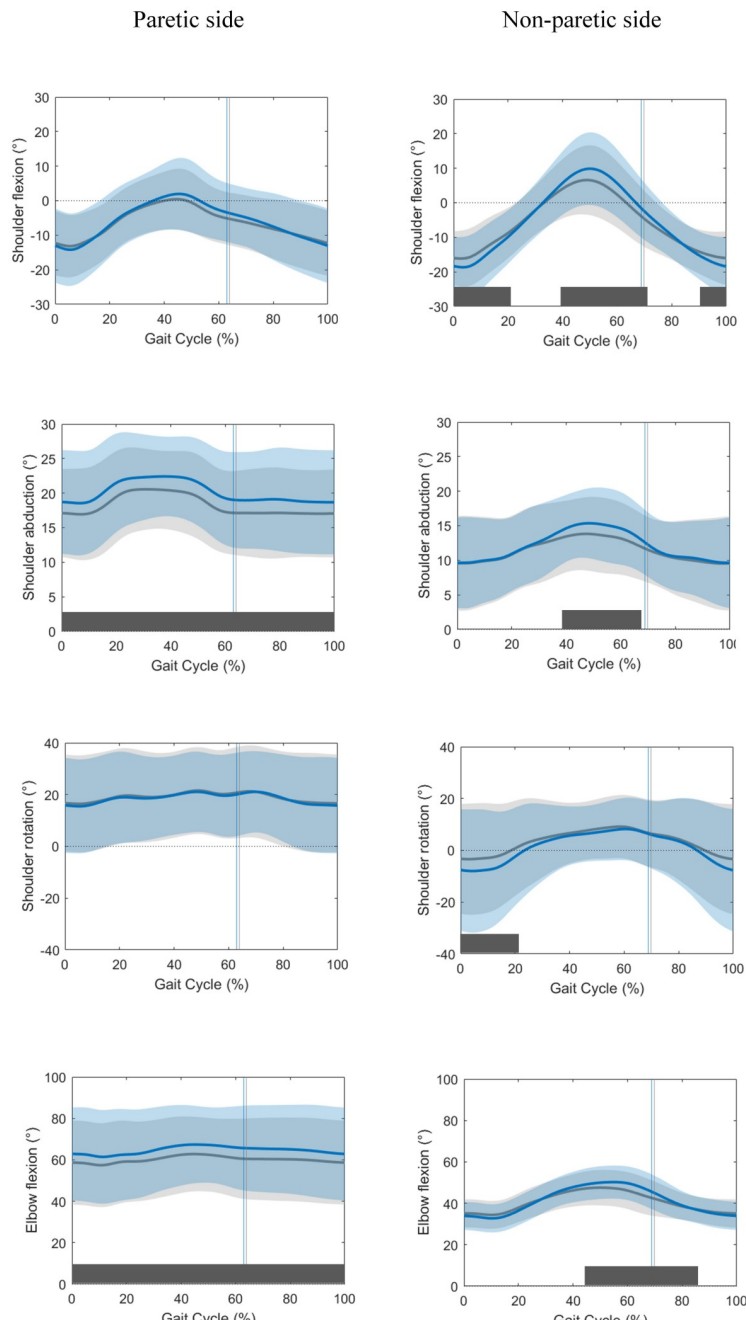

**Fig 2. Kinematic curves for the arm swing at the paretic and non-paretic side in both conditions.** Average kinematic curves and standard deviations of the CWS condition (grey) and FWS condition (blue). Positive results in the y-axis represent: shoulder flexion,—abduction,—internal rotation and elbow flexion. The vertical lines indicate the moment of push-off in each condition. Parts of the gait cycle where significant differences between CWS and FWS were detected using SPM analysis are indicated by dark grey bars.

significant speed-dependent changes were only present at the non-paretic side with increases in shoulder extension (0–21% GC, p = 0.007; 91–100% GC, p = 0.034), peak shoulder flexion (40–75% GC, p<0.001) and elbow flexion (39–75% GC, p<0.001). Speed-dependent increases in shoulder abduction (38–68% GC, p = 0.001) and external rotation (0–23% GC, p = 0.004)

were also found at the non-paretic side. At the level of the trunk, only a significantly (p = 0.044) increased trunk forward flexion was present (58–72% GC) in FWS compared to CWS (S2 Fig in S1 File).

Shoulder and elbow ROM and mean elbow flexion values can be found in Table 2. Individual results for sagittal plane shoulder ROM are displayed in S3 Fig in S1 File. Although no

**Table 2. Comparison of the active range of motion (°) of the shoulder and elbow (paretic side and non-paretic side) in the entire group and for the subgroups.**

| | CWS | FWS | MEAN DIFFERENCE (SD) | P |
|---|---|---|---|---|
| | Mean (SD) | Mean (SD) | | |
| **Paretic side** | | | | |
| **Full group (N = 25)** | | | | |
| Sh Sagittal plane (range) | 16.49 (9.2) | 19.31 (11.6) | 2.82 (4.0) | **< 0.001**\* |
| Sh Frontal plane (range) | 6.24 (3.0) | 7.19 (4.3) | 0.96 (2.3) | **0.009**\* |
| Sh Transverse plane (range) | 10.08 (5.2) | 11.00 (5.3) | 0.92 (2.8) | 0.122 |
| El Sagittal plane (range) | 8.78 (6.1) | 10.12 (7.5) | 1.34 (2.6) | **0.007**\* |
| El Sagittal plane (mean) | 60.13 (19.6) | 64.61 (21.2) | 4.48 (4.4) | **< 0.001**\* |
| **1:1 (N = 15)** | | | | |
| Sh Sagittal plane (range) | 21.14 (8.9) | 24.50 (12.1) | 3.36 (4.9) | **0.006**\* |
| Sh Frontal plane (range) | 5.98 (3.5) | 7.31 (5.3) | 1.33 (2.6) | **0.008**\* |
| Sh Transverse plane (range) | 10.36 (6.0) | 11.57 (6.6) | 1.21 (2.8) | 0.191 |
| El Sagittal plane (range) | 11.03 (6.6) | 13.33 (8.0) | 2.30 (2.3) | **0.003**\* |
| El Sagittal plane (mean) | 54.21 (16.1) | 57.31 (17.0) | 3.10 (3.9) | **0.006**\* |
| **2:1 (N = 9)** | | | | |
| Sh Sagittal plane (range) | 9.03 (3.7) | 11.19 (4.1) | 2.16 (2.2) | **0.028**\* |
| Sh Frontal plane (range) | 6.84 (2.1) | 7.28 (2.1) | 0.44 (1.7) | 0.441 |
| Sh Transverse plane (range) | 9.69 (4.2) | 10.20 (2.5) | 0.51 (3.1) | 0.441 |
| El Sagittal plane (range) | 4.95 (2.7) | 5.03 (2.2) | 0.08 (2.5) | 0.441 |
| El Sagittal plane (mean) | 72.53 (20.0) | 78.97 (21.5) | 6.44 (4.8) | **0.011**\* |
| **Non-paretic side** | | | | |
| **Full group (N = 25)** | | | | |
| Sh Sagittal plane (range) | 24.64 (12.3) | 29.94 (13.1) | 5.30 (3.5) | **< 0.001**\* |
| Sh Frontal plane (range) | 6.29 (4.0) | 7.80 (3.9) | 1.51 (2.3) | **< 0.001**\* |
| Sh Transverse plane (range) | 17.53 (13.1) | 21.62 (15.5) | 4.09 (5.8) | **< 0.001**\* |
| El Sagittal plane (range) | 15.47 (7.4) | 19.56 (7.7) | 4.09 (2.7) | **< 0.001**\* |
| El Sagittal plane (mean) | 40.64 (6.5) | 41.24 (6.2) | 0.60 (2.1) | 0.242 |
| **1:1 (N = 15)** | | | | |
| SH Sagittal plane (range) | 25.89 (13.8) | 30.87 (14.5) | 4.98 (3.1) | **< 0.001**\* |
| SH Frontal plane (range) | 6.20 (4.7) | 7.49 (4.0) | 1.29 (1.7) | **0.008**\* |
| SH Transverse plane (range) | 14.58 (8.0) | 17.35 (9.3) | 2.77 (2.9) | **0.002**\* |
| EL Sagittal plane (range) | 16.87 (8.0) | 21.43 (7.8) | 4.56 (2.9) | **< 0.001**\* |
| EL Sagittal plane (mean) | 42.91 (6.6) | 42.83 (6.1) | 0.08 (1.7) | 0.910 |
| **2:1 (N = 9)** | | | | |
| Sh Sagittal plane (range) | 21.48 (9.9) | 27.81 (11.8) | 6.33 (4.2) | **0.008**\* |
| Sh Frontal plane (range) | 6.84 (2.8) | 8.78 (3.6) | 1.94 (2.4) | 0.051 |
| Sh Transverse plane (range) | 18.52 (15.2) | 25.76 (20.8) | 7.24 (8.0) | **0.008**\* |
| El Sagittal plane (range) | 13.06 (6.4) | 16.90 (7.3) | 3.84 (1.8) | **0.008**\* |
| El Sagittal plane (mean) | 37.62 (5.2) | 39.39 (6.1) | 1.77 (2.4) | 0.066 |

CWS = comfortable walking speed; FWS = fast walking speed; Sh = shoulder; El = elbow; Wilcoxon Signed Rank test;

\* p < 0.05

significant difference for the sagittal shoulder movements at the paretic side was present between the two conditions based on SPM comparison, a significant speed-dependent increase in ROM for shoulder flexion/extension (2.82 ± 4.0˚, p<0.001) was found. ROM also increased for all other shoulder and elbow movements except for the paretic shoulder rotations (Table 2).

Correlation coefficient values between the speed-dependent changes in upper limb ROM and the clinical and gait parameters for the entire group can be found in Fig 4. Participants with a higher score on the FMUL and FMLL, indicating a less impaired upper (r = 0.600, p = 0.002) and lower limb (r = 0.416, p = 0.039), a smaller step width at CWS (r = -0.400, p = 0.048) and less spasticity in the internal shoulder rotators (r = -0.503, p = 0.010) and elbow flexors (r = -0.422, p = 0.035) showed greater speed-dependent changes in ROM for elbow flexion. Persons with a lower score in the FMUL and thus a more impaired upper limb (r = -0.521, p = 0.008) and a wider step width at FWS (r = 0.534, p = 0.006) showed a speed-dependent increase in mean elbow flexion.

## Speed-dependent changes in arm swing for subgroups

The upper limb kinematic curves of participants in the 1:1 and 2:1 subgroups are presented in Fig 3. Walking speed did not differ between subgroups. Participants in the *1:1 subgroup (N = 15)* showed increased shoulder abduction (16–86% GC, p<0.001) and elbow flexion (46–86% GC, p = 0.001) at the paretic side at FWS compared to CWS. At the non-paretic side, they showed increased shoulder extension (0–18% GC, p = 0.012; 92–100% GC, p = 0.038) and peak shoulder flexion (45–67% GC, p = 0.006). Additionally, increased shoulder abduction

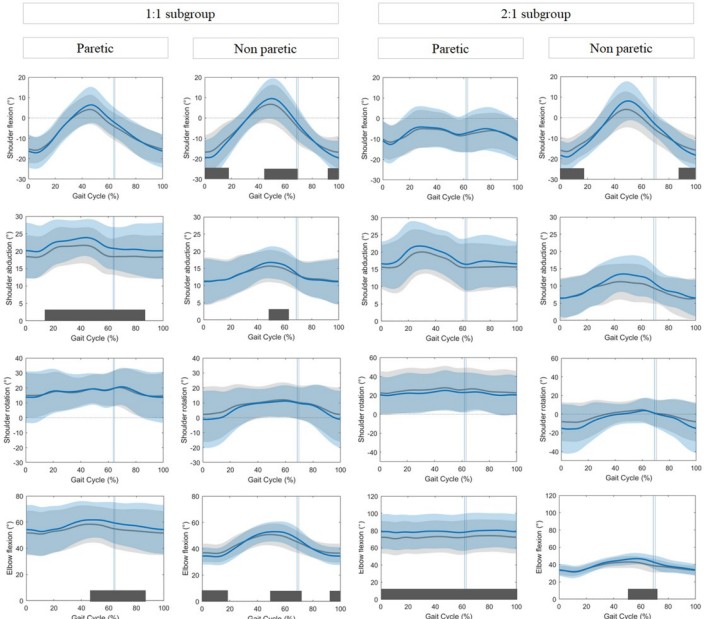

**Fig 3. Kinematic curves for the arm swing at the paretic and non-paretic side for both subgroups.** Average kinematic curves and standard deviations of the CWS condition (grey) and FWS condition (blue). The two left columns represent the 1:1 subgroup, the two right columns the 2:1 subgroup. Positive results in the y-axis represent following movements: shoulder flexion,—abduction,—internal rotation and elbow flexion. Parts of the gait cycle where significant differences were detected are indicated by dark grey bars. The vertical lines indicate the moment of push-off in each condition.

(51–62% GC, p = 0.027), elbow extension (0–19% GC, p = 0.004; 92–100% GC, p = 0.032) and elbow flexion (49–72% GC, p = 0.032) was found.

Participants assigned to the *2:1 subgroup (N = 9)* showed significantly greater paretic elbow flexion (p<0.001) at FWS over the entire gait cycle. At the non-paretic side, a speed-dependent increase in shoulder extension (0–17% GC, p = 0.001, 89–100% GC, p = 0.007) and elbow flexion (50.1–73.1% GC, p<0.001) was found. In contrast to the other group, no significant difference was present for the peak shoulder flexion in the middle part of the gait cycle. No speed-dependent significant differences could be detected for trunk movements in both subgroups.

Both subgroups showed a significant increase in almost all shoulder and elbow ROM values at the non-paretic side during FWS compared to CWS (Table 2). At the paretic side, participants of the 1:1 subgroup showed an increase in all ROM parameters except for the shoulder abduction. Participants of the 2:1 subgroup only showed an increase in shoulder flexion ROM (p = 0.028) and mean elbow flexion (p = 0.011).

Correlation coefficient values between the speed-dependent changes in upper limb ROM and clinical and gait parameters for both subgroups can be found in Fig 4. Participants of the 1:1 subgroup with a lower score on the FMUL and thus a more impaired upper limb (r = -0.561, p = 0.030) or a wider step width (CWS r = 0.604, p = 0.017; FWS r = 0.721, p = 0.002) showed a speed-dependent increase in mean elbow flexion. Participants with a higher score on the FMUL, implicating a less impaired upper limb (r = 0.634, p = 0.011) or less spasticity in the shoulder internal rotators (r = -0.604, p = 0.017) showed a speed-dependent increase in elbow ROM. Participants of the 2:1 subgroup with a lower score on the FMUL and thus a more impaired lower limb (r = -0.828, p = 0.006), wider step width (CWS r = 0.817, p = 0.007; FWS r = 0.833, p = 0.005) or less spasticity in the shoulder internal rotators (r = -0.693, p = 0.038) showed greater speed-dependent changes for shoulder rotation ROM (Fig 4).

## Discussion

Addressing the first aim of this study, to examine changes in upper limb kinematics with increasing walking speed, the current results revealed that the non-paretic upper limb adapts to a faster walking speed by increasing the ROM of the shoulder in all the three planes and of the elbow, in the sagittal plane. On the other hand, the paretic upper-limb only shows changes in shoulder abduction and elbow flexion. These findings support our hypothesis that upper limb ROM increases with walking speed partly. While there was an overall increased shoulder flexion ROM at the paretic side with increasing walking speeds, which was smaller compared to the non-paretic side, the responses were heterogenous. For the second aim, which explores whether the speed-dependent changes in the upper limb differ depending on the coordination pattern of the paretic arm to the non-paretic leg (arm-to-leg swing ratio), the subgroup analysis revealed that persons post-stroke with a more impaired arm swing (i.e. 2:1 subgroup) showed less speed-dependent adaptations and a greater increase of compensatory mean elbow flexion over the entire gait cycle. These findings imply that an altered arm-to-leg swing ratio post-stroke limits the potential to adapt arm swing movements, thus confirming the hypothesis that individuals with a more impaired arm swing have less ability to adjust their movements at higher walking speeds.

There was some discrepancy between the results of the SPM analysis and the comparison of discrete parameters as for example, the shoulder flexion. The ROM for the discrete parameters was calculated as the difference between the minimum and maximum amount of shoulder flexion over the entire gait cycle without considering the timing over the gait cycle. When inspecting the individual graphs of all participants (S4 Fig in S1 File) we observed that the

| | FMUL | FMLL | Δ walking speed | Step width CWS | Step width FWS | Spasticity Internal Rotators Shoulder | Spasticity Elbow Flexors |
|---|---|---|---|---|---|---|---|
| **ENTIRE GROUP (N=25)** | | | | | | | |
| Δ Sh. sagittal plane (ROM) | 0.110 | -0.147 | 0.221 | -0.078 | -0.182 | -0.062 | -0.021 |
| Δ Sh. frontal plane (ROM) | 0.170 | -0.211 | 0.235 | 0.128 | 0.073 | -0.197 | 0.017 |
| Δ Sh. transverse plane (ROM) | -0.176 | -0.098 | -0.125 | -0.025 | -0.149 | 0.203 | 0.200 |
| Δ El. sagittal plane (ROM) | **0.600**\*\* | **0.416**\* | 0.305 | **-0.400**\* | -0.345 | **-0.503**\* | **-0.422**\* |
| Δ El. sagittal plane (Mean) | **-0.521**\*\* | -0.191 | 0.123 | 0.347 | **0.534**\*\* | 0.326 | 0.374 |
| **1:1 SUBGROUP (N=15)** | | | | | | | |
| Δ Sh. sagittal plane (ROM) | 0.244 | -0.089 | 0.296 | -0.279 | -0.396 | -0.008 | -0.217 |
| Δ Sh. frontal plane (ROM) | 0.394 | -0.224 | 0.147 | 0.036 | 0.157 | -0.483 | 0.031 |
| Δ Sh. transverse plane (ROM) | -0.093 | 0.244 | 0.000 | -0.400 | -0.464 | 0.003 | 0.247 |
| Δ El. sagittal plane (ROM) | **0.634**\* | 0.126 | 0.496 | -0.482 | -0.396 | **-0.604**\* | -0.464 |
| Δ El. sagittal plane (Mean) | **-0.561**\* | 0.071 | 0.236 | **0.604**\* | **0.721**\*\* | 0.090 | 0.124 |
| **2:1 SUBGROUP (N=9)** | | | | | | | |
| Δ Sh. sagittal plane (ROM) | -0.200 | -0.603 | 0.119 | 0.567 | 0.617 | 0.286 | 0.239 |
| Δ Sh. frontal plane (ROM) | -0.267 | -0.243 | 0.186 | 0.433 | 0.367 | 0.217 | 0.100 |
| Δ Sh. transverse plane (ROM) | -0.617 | **-0.828**\*\* | -0.186 | **0.817**\*\* | **0.833**\*\* | **-0.693**\* | 0.249 |
| Δ El. sagittal plane (ROM) | 0.400 | 0.460 | -0.085 | -0.100 | 0.033 | -0.217 | -0.398 |
| Δ El. sagittal plane (Mean) | 0.183 | -0.109 | 0.254 | -0.133 | -0.033 | 0.156 | 0.438 |

This is the Table 3. Legend
N = number; FMUL = Fugl-Meyer Upper Limb; FMLL = Fugl-Meyer Lower Limb; CWS = comfortable walking speed; FWS = fast walking speed; Sh. = shoulder; El. = elbow

Spearman correlation coefficients
**\*\* Correlation is significant at the 0.01 level (2-tailed)**
**\* Correlation is significant at the 0.05 level (2-tailed)**

**Fig 4. Correlations between the difference (Δ) in upper limb kinematics between comfortable (CWS) and fast (FWS) walking speed conditions and the clinical and gait parameters.**

timing of maximum shoulder flexion was very heterogenous. Since whole trajectory analysis such as SPM accounts for differences in both the amplitude as the timing, this may explain why no difference was detected in shoulder flexion ROM using SPM, but did reveal a significant difference as a discrete parameter. This discrepancy highlights the advantages of applying

both approaches to fully understand the kinematic and temporal differences due to increasing walking velocity [31].

Finally, in addressing the third aim, to further explain the observed heterogeneity, correlation coefficients were calculated between speed-dependent changes in upper limb ROM and upper/lower limb impairment (FMUL, FMLL), the increase in walking speed, step width during comfortable and fast walking and spasticity of the internal shoulder rotators and elbow flexors. Since an increase in arm swing amplitude is probably initiated by an increased arm muscle activity [8] in addition to potential changes in the passive mechanism of the arm swing, we expected more speed-dependent changes in persons with higher scores on the Fugl-Meyer assessment. However, only correlations between speed-dependent changes in elbow ROM and Fugl-Meyer scores, step width during the CWS condition and spasticity of the elbow flexors were found. Persons with less impairment of the upper and lower limb (lower scores in FMUL and FMLL), a smaller step width during CWS and less spasticity in the shoulder internal rotators or elbow flexors showed a larger increase in elbow flexion ROM during fast walking. Additionally, persons with a more impaired upper limb (lower score on FMUL) and a wider step width during FWS showed an increased average elbow flexion position during fast walking. This increased elbow flexion is in contrast to healthy controls who show an increased elbow flexion ROM with a phasic movement pattern when increasing walking speed [7] and has been described earlier as a potential associated reaction or increase in muscle tone in persons with acquired brain injury or stroke. Since these associated reactions during walking cannot be attributed to a single contributing factor [32], this probably explains why only a moderate correlation was detected. Similarly, in the subgroups, the detected correlations were mostly related to speed-dependent changes in elbow movements. Only participants of the 2:1 subgroup with a more impaired lower limb (lower score on FMLL) and wider step width showed greater speed-dependent changes for shoulder rotations.

Participants in the current study could increase their walking speed with a velocity change that was higher than the minimal clinical important difference, which is in line with previous reports on independent walkers post stroke [26]. Despite the noticed increase in shoulder ROM, most changes occurred at the elbow and are related to impairment level and walking stability (i.e. step width). This is in line with earlier research that showed that the increase in walking speed is limited by balance impairments in persons post-stroke [33]. Therefore, future studies investigating therapeutic interventions that target the arm swing, should focus on both the paresis and walking stability to control the position and movements of the elbow in addition to focusing on better shoulder movements. Afterall, earlier studies already showed improved gait parameters with the arm positioned in a more extended position after treatment with botulinum toxin A [16–18].

## Limitations

Current results are only applicable to high-functioning persons post-stroke able to walk independently on a treadmill. Especially the sample size of the 2:1 subgroup was very small (N = 9) and the group heterogenous. The gait pattern and associated arm swing could be related to the time since stroke. Although current study population covers a wide range of time post-stroke, no significant difference was detected for this variable between subgroups. Experience with treadmill walking was not considered yet the familiarization period of six minutes provided to all participants helped avoid any potential influences. No causal relationships were investigated as only correlation statistics was performed in this study. Finally, future research should also measure muscle activity to investigate the motor control underlying the kinematics that are

presented in current results and whether improvements in upper-limb kinematics as part of a rehabilitation program results in increased walking speed.

## Conclusion

Independent walkers post-stroke show different changes in arm swing kinematics at the paretic side compared to the non-paretic side when increasing walking speed. Since persons with greater upper-limb impairment and a reduced stability during walking show more compensatory elbow movements, future studies investigating therapeutic interventions aiming to improve the arm swing during walking should target both contributing factors.

## Supporting information

**S1 File.**
(DOCX)

**S2 File.**
(ZIP)

## Author Contributions

**Conceptualization:** Dirk Cambier, Anke Van Bladel.

**Data curation:** Anke Van Bladel.

**Formal analysis:** Daan De Vlieger, Arne Defour, Lynn Bar-On, Anke Van Bladel.

**Methodology:** Dirk Cambier, Anke Van Bladel.

**Visualization:** Daan De Vlieger, Arne Defour, Lynn Bar-On, Anke Van Bladel.

**Writing – original draft:** Daan De Vlieger, Arne Defour, Lynn Bar-On, Anke Van Bladel.

**Writing – review & editing:** Daan De Vlieger, Arne Defour, Lynn Bar-On, Dirk Cambier, Eva Swinnen, Ruth Van der Looven, Anke Van Bladel.

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
