## [Decision Letter · Decision Letter 0]

27 Sep 2024

PONE-D-24-22190Speed-dependent changes in the arm swing during independent walking in individuals after stroke.PLOS ONE

Dear Dr. Van Bladel,

Thank you for submitting your manuscript to PLOS ONE. After careful consideration, we feel that it has merit but does not fully meet PLOS ONE’s publication criteria as it currently stands. Therefore, we invite you to submit a revised version of the manuscript that addresses the points raised during the review process.

**ACADEMIC EDITOR:**Dear Authors,your manuscript had been revised by one expert in the field that reported some minor issues you should consider during the revision process.

We look forward to receiving your revised manuscript.

Kind regards,

Emiliano Cè

Academic Editor

PLOS ONE

Journal Requirements:

2. Please include a caption for figure 1, 2 and 3.

Reviewers' comments:

Reviewer's Responses to Questions

**Comments to the Author**

1. Is the manuscript technically sound, and do the data support the conclusions?

Reviewer #1: Partly

2. Has the statistical analysis been performed appropriately and rigorously? 

Reviewer #1: Yes

3. Have the authors made all data underlying the findings in their manuscript fully available?

Reviewer #1: Yes

4. Is the manuscript presented in an intelligible fashion and written in standard English?

Reviewer #1: Yes

5. Review Comments to the Author

Reviewer #1: Date: 9/25/2024

Subject: Reviewer Comments

“Speed-dependent changes in the arm swing during independent walking in individuals after stroke.”

Manuscript ID: PONE-D-24-22190

Overall Summary

The purpose of this study was to describe differences in upper extremity kinematics in response to speed changes for persons with stroke. A secondary purpose was to test for differences in subgroups based on 1:1 or 2:1 arm swing pattern.

In general, the authors used appropriate research design, analytical methods, and statistical analyses. The manuscript was clearly written, and the figures presented data visualization well. I have some comments for the authors to consider.

Major Concerns

1. The authors claim that there is a causal relationship between walking speed and upper extremity movements to support rationale for treatment. For example see line 96. However, the authors do cite supporting evidence to demonstrate a causal relationship, nor does the current study design provide evidence for causality as it is a descriptive, cross-sectional design. I would caution the authors about using statements that suggest causality in the introduction and conclusion.

2. The authors do not discuss their findings with respect to passive mechanical coupling that can occur as a result of increased speed. For example, increases in upper extremity ROM can be the result of mechanical transfer of ground reaction forces through the skeletal, ligamentous, and passive properties of muscle tissue rather than active muscle contractile properties. Adding this possible hypothesis to the discussion will provide a more comprehensive picture for readers. Additionally, the current study design does not differentiate whether the observed changes are a result of impairment or an artifact of increased speed. The use of the term speed-dependency would imply that the resultant changes are not influenced by impairment. Further studies will be needed to elucidate these differences.

3. The discussion could be strengthened by addressing explicitly whether the findings supported or refuted the proposed hypotheses.

Minor Concerns

Abstract:

Line 49. This sentence appears to be incomplete?

Introduction:

Line 103-105. The introduction would be strengthened for readers if these subgroup patterns were introduced prior to the study purpose.

Results:

Line 250-254. Please consider re-wording this section to clarify if these are descriptive characteristics of the group or relationships between variables within the subgroups. I had to return to tables to try and understand the clinical descriptives for each subgroup.

Discussion:

Line 276-277. Please consider deleting as this statement does not describe how the findings support or refute the hypotheses.

Limitations:

Please add that these results may not be generalizable to all persons post-stroke provided the high functional level based on gait speed.

6. PLOS authors have the option to publish the peer review history of their article (what does this mean?). If published, this will include your full peer review and any attached files.

Reviewer #1: No

---

## [Author Response · Author response to Decision Letter 0]

28 Oct 2024

“Speed-dependent changes in the arm swing during independent walking in individuals after stroke.”

Manuscript ID: PONE-D-24-22190

Dear editor and reviewer,

Thank you for the valuable feedback and the time and effort that you have put in the review of this manuscript. We addressed your comments and questions in the hope this will meet your expectations. We believe that in this way the quality of the manuscript has improved accordingly.

Major Concerns

1. The authors claim that there is a causal relationship between walking speed and upper extremity movements to support rationale for treatment. For example, see line 96. However, the authors do cite supporting evidence to demonstrate a causal relationship, nor does the current study design provide evidence for causality as it is a descriptive, cross-sectional design. I would caution the authors about using statements that suggest causality in the introduction and conclusion.

Thank you for this remark. We agree that the assumption that rehabilitation based changes in upper limb movements may help to increase walking speed in subjects post-stroke is hypothetical and based only on rationale from studies performed in healthy individuals (see lines 66-70 in the introduction). Moreover, we agree that the study designs of the referenced papers that have been carried out in the stroke populations should have been considered and that the results of the current study cannot indicate a causal relationship. To avoid causality statements and to make it clearer that our rationale for possible implications for rehabilitation are hypothetical, we have made the following amendments:

Abstract:

Line 55-56 original

Persons post-stroke show different speed-dependent changes in the paretic arm swing compared to the non-paretic side.

Line 55-56 adjusted

Persons post-stroke show different changes in arm swing kinematics at the paretic compared to the non-paretic side when increasing walking speed.

Introduction:

Line 96-97 original

Knowing whether persons post-stroke are capable of adapting their arm movements can aid post-stroke gait rehabilitation aiming to increase walking speed.

Line 96-97 adjusted

Knowing whether persons post-stroke are capable of adapting their arm movements, and which factors are associated with this ability, may be a first step in aiding post-stroke gait rehabilitation as a potential way to contribute to walking speed.

Discussion:

Line 350-353 added

No causal relationships were investigated as only correlation statistics was performed in this study. Finally, future research should also measure muscle activity to investigate the motor control underlying the kinematics that are presented in current results and whether improvements in upper-limb kinematics as part of a rehabilitation program results in increased walking speed.

Conclusions:

Line 355-356 original

Independent walkers post-stroke show different speed-dependent arm swing changes at the non-paretic and paretic side when increasing walking speed.

Line 355-356 adjusted

Independent walkers post-stroke show different changes in arm swing kinematics at the paretic compared to the non-paretic side when increasing walking speed.

2. The authors do not discuss their findings with respect to passive mechanical coupling that can occur as a result of increased speed. For example, increases in upper extremity ROM can be the result of mechanical transfer of ground reaction forces through the skeletal, ligamentous, and passive properties of muscle tissue rather than active muscle contractile properties. Adding this possible hypothesis to the discussion will provide a more comprehensive picture for readers. Additionally, the current study design does not differentiate whether the observed changes are a result of impairment or an artifact of increased speed. The use of the term speed-dependency would imply that the resultant changes are not influenced by impairment. Further studies will be needed to elucidate these differences.

To be more complete, we have added information to the revised discussion about the potential contribution of the passive mechanisms of arm swing in the discussion (line 312-313). However, based on the paper of Goudriaan et al. (2014) we have reasons to believe that an increase in muscle activity is necessary to increase the arm swing range of motion during fast walking. In their study, Goudriaan and colleagues simulated the arm swing amplitude in different walking speeds without adding upper limb muscle activity to their simulation model. Their results showed that the range of motion remained consistent across different walking speeds when walking without upper limb muscle activation. Subsequently, they concluded that increased arm swing amplitude requires muscle activation. 

Original

Since an increase in arm swing amplitude is probably initiated by an increased arm muscle activity (8), we expected more speed-dependent changes in persons with higher scores on the Fugl-Meyer assessment.

Line 315-317 adjusted

“Since an increase in arm swing amplitude is probably initiated by an increased arm muscle activity (8) in addition to potential changes in the passive mechanism of the arm swing, we expected more speed-dependent changes in persons with higher scores on the Fugl-Meyer assessment.”

Related to the second part of your concern the intention was not to investigate causal relationships in this study. We used correlation statistics to investigate which outcome measures potentially contribute to the observed changes. We acknowledge the limitations of this statistical approach and have therefore tried to clarify this in lines 349-350 of the revised discussion.

“No causal relationships were investigated as only correlation statistics was performed in this study. Finally, future research should also measure muscle activity to investigate the motor control underlying the kinematics that are presented in current results and whether improvements in upper-limb kinematics as part of a rehabilitation program results in increased walking speed.”

Finally, we understand your concerns about the term “speed-dependent changes”. However, since the different conditions compared in this cross-sectional study were performed at the same time and the only difference between these conditions was the walking speed, we believe that the changes described in the results section are related to this change in walking speed and not to changes in impairment level. The authors do not intend to suggest that only the change in speed contributes to the observed kinematics. Other factors, like impairment, probably also contribute to the observed kinematics. However, these did not change between the assessment of the two conditions and this research question was not addressed in the current study.

3. The discussion could be strengthened by addressing explicitly whether the findings supported or refuted the proposed hypotheses.

Thank you for this suggestion. We have added this information to the discussion (lines 286-300).

“Addressing the first aim of this study, to examine changes in upper limb kinematics with increasing walking speed, the current results revealed that the non-paretic upper limb adapts to a faster walking speed by increasing the ROM of the shoulder in all the three planes and of the elbow, in the sagittal plane. On the other hand, the paretic upper limb only shows changes in shoulder abduction and elbow flexion. These findings support our hypothesis that upper limb ROM partly increases with walking speed. While there was an overall increased shoulder flexion ROM at the paretic side with increasing walking speeds, which was smaller compared to the non-paretic side, the responses were heterogenous. For the second aim, which explored whether the speed-dependent changes in the upper limbs differ depending on the coordination pattern of the paretic arm (arm-to-leg swing ratio), the subgroup analyses revealed that persons post-stroke with a more impaired arm swing (i.e. 2:1 subgroup) showed less speed-dependent adaptations and a greater increase of compensatory mean elbow flexion over the entire gait cycle. These findings imply that an altered arm-to-leg swing ratio post-stroke limits the potential to adapt arm swing movements, thus confirming the hypothesis that individuals with more impaired arm swing have less ability to adjust their movements at higher walking speeds.”

Minor Concerns

Abstract: 

Line 49. This sentence appears to be incomplete?

Thank you for the remark. We have revised the sentence accordingly.

“Persons post-stroke with a more impaired arm swing coordination pattern only showed speed-dependent adaptations for elbow flexion (p<0.001) at the paretic side during fast walking.”

Introduction:

Line 103-105. The introduction would be strengthened for readers if these subgroup patterns were introduced prior to the study purpose.

We introduced the subgroups (based on the papers of Wagenaar et al, 1994 and Van Bladel et al, 2023) prior to the study purposes by placing the sentence below earlier in lines 81-82.

“While in healthy adults a 2:1 arm-to-leg swing ratio (two arm swings, one stride) can occur when walking slower than the comfortable walking speed, persons post-stroke sometimes show a 2:1 ratio on the paretic side independent of walking speed. Wagenaar et al. (1994) previously described these two arm-swing coordination patterns (9), which were recently confirmed by Van Bladel et al. (2023) (21).”

Results:

Line 250-254. Please consider re-wording this section to clarify if these are descriptive characteristics of the group or relationships between variables within the subgroups. I had to return to tables to try and understand the clinical descriptives for each subgroup.

We have added a sentence that links this specific part to Table 3, which shows the correlations between variables within the subgroups (now lines 256-257).

“Correlation coefficient values between the speed-dependent changes in upper limb ROM and clinical and gait parameters for both subgroups can be found in table 3.”

We clarified the clinical description of the degree of impairment in relation to the score on the FMUL in the results (lines 228-234, 255-263) and throughout the discussion.

Discussion:

Line 276-277. Please consider deleting as this statement does not describe how the findings support or refute the hypotheses.

Thank you for your remark, this statement has been removed.

Limitations:

Please add that these results may not be generalizable to all persons post-stroke provided the high functional level based on gait speed.

This has been added at line 344.

“Current results are only applicable to high-functioning persons post-stroke able to walk independently on a treadmill. “

Note to the editor:

We have recently received financial support from the Universitaire Stichting van België to pay the publication fee.

Therefore, we want to add this to the funding section of the paper.

"Published with support of the Universitaire Stichting from Belgium and of the Rehabilitation technology for persons with a Brain Injury alliance research group”

---

## [Decision Letter · Decision Letter 1]

25 Nov 2024

Speed-dependent changes in the arm swing during independent walking in individuals after stroke.

PONE-D-24-22190R1

Dear Dr. Van Bladel,

We’re pleased to inform you that your manuscript has been judged scientifically suitable for publication and will be formally accepted for publication once it meets all outstanding technical requirements.

Kind regards,

Emiliano Cè, Ph.D.

Academic Editor

PLOS ONE

Additional Editor Comments (optional):

Reviewers' comments:

Reviewer's Responses to Questions

**Comments to the Author**

1. If the authors have adequately addressed your comments raised in a previous round of review and you feel that this manuscript is now acceptable for publication, you may indicate that here to bypass the “Comments to the Author” section, enter your conflict of interest statement in the “Confidential to Editor” section, and submit your "Accept" recommendation.

Reviewer #1: All comments have been addressed

2. Is the manuscript technically sound, and do the data support the conclusions?

Reviewer #1: Yes

3. Has the statistical analysis been performed appropriately and rigorously? 

Reviewer #1: Yes

4. Have the authors made all data underlying the findings in their manuscript fully available?

Reviewer #1: Yes

5. Is the manuscript presented in an intelligible fashion and written in standard English?

Reviewer #1: Yes

6. Review Comments to the Author

Reviewer #1: Thank you for your detailed explanations and revisions. Congratulations on completing this excellent research study. It was a pleasure to review and I do not have any more recommendations.

7. PLOS authors have the option to publish the peer review history of their article (what does this mean?). If published, this will include your full peer review and any attached files.

Reviewer #1: No

---

## [Editor Report · Acceptance letter]

29 Nov 2024

PONE-D-24-22190R1 

PLOS ONE

Dear Dr. Van Bladel, 

I'm pleased to inform you that your manuscript has been deemed suitable for publication in PLOS ONE. Congratulations! Your manuscript is now being handed over to our production team.

Kind regards, 

on behalf of

Prof. Emiliano Cè 

Academic Editor

PLOS ONE